

# Characterization of Ocean Mixing and Dynamics during the 2017 Maud Rise Polynya Event

Jhon F. Mojica[1], Daiane Faller[1], Diana Francis[1], Clare Eayrs[1], David Holland[2]

[1] Center for global Sea Level Change, NYUAD, Abu Dhabi, UAE.

[2] Courant Institute of Mathematical Sciences, NYUNY, New York, USA.

*Correspondence to*: Jhon F. Mojica (jhon.mojica@nyu.edu)

**Abstract.** During 2017 Austral winter, an open ocean polynya appeared in the Lazarev Sea, centered over Maud Rise. The vertical structure of the water column presented temporal and spatial variability with a weak stratification during the period of observations from January
2015 to January 2019. While over the Northern Maud Rise area, a highly stratified layer was identified between 80 – 180 m depth. This layer works as a thermal barrier where the energy from summer months is stored, preventing the warm sub-surface waters from mixing with the shallow waters. So far, a complete description of the polynya formation and maintenance processes is still lacking. To characterize the internal structure of the ocean during the 2017
Maud Rise polynya event we use *in situ* observations, and ocean model reanalysis data. The obtained results revealed that the incidence of thermobaric convection, diapycnal and isopycnal mixing processes over the Maud Rise drives the exchange of energy in the water column. We highlight three relevant factors that contribute to the energy flux for the open-ocean polynya preconditioning: level of instability, pycnocline fluctuation, and bathymetric influence.
Another remarkable feature is the warmer summer surface layer over the Maud Rise, which transfers heat to intermediate layers accumulating energy for almost four months. Energy storage at the thermal barrier is evaluated based on heat flux calculations to quantify the exchange of energy between the different water layers. These processes together operate as an ocean preconditioning to the formation and maintenance of an open-ocean polynya event.

**Key Words.** Maud Rise Polynya, mixing rates, thermal barrier, sea ice, heat fluxes, convection, Lazarev Sea, Antarctica

## 1. Introduction

The Weddell Sea (WS) is characterized by a coastal westward flowing current around Antarctica that is driven by the prevailing atmospheric easterlies and thermohaline forcing (Tuner and Marshall, 2011). This current forms part of a cyclonic (clockwise) circulation, the so-called Weddell Gyre (WG), which is bounded by the Antarctic coast in the south, the
Antarctic Peninsula in the west, and the Scotia Ridge in the north (Orsi et al., 1993). Such cyclonic gyre features and patterns are observed in other Antarctic regions, such as the Ross Sea (Dotto et al., 2018). This circulation produces an upwelling due to large-scale overturning that increases vertical exchange and promotes the renewal of deep water, an essential component to determine the regional sea ice extent (Gordon, 1982, Nicholls et al., 2009).



Cold and fresh Antarctic Surface Waters (ASW) overlay the warm and salty Weddell Deep
        Water (WDW, 200 –1500 m, Carmack, 1974). WDW is supplied by the Circumpolar Deep
        Water (CDW), which diverges from the Antarctic Circumpolar Current (ACC, Gordon, 1982).
        CDW is limited to south of 65° S and has a transport of ~50 Sv (Fahrbach and Beckmann,
2001). Meanwhile, below these waters are located the Weddell Sea Deep Water (WSDW) with
        a characteristic temperature range 0 to -0.6 °C and the Weddell Sea Bottom Water (WSBW)
        with temperatures < - 0.6 °C. The WSDW and WSBW sinking contribute to a colder mass,
        known as Antarctic Bottom Water (AABW) that is an essential factor to ocean currents and
        circulation around the world (Deacon, 1963).

        Freshwater fluxes produced by the freezing and melting sea ice cycle are a critical component
        of the ocean circulation over the WS. The sea ice movements do not only depend on the surface
        currents; the sea ice drift patterns also reflect the wind surface velocity field. The WS area is
        characterized by a weak stratification that implies a strong barotropic component of the currents
(Fahrbach and Beckmann, 2001). The ocean waters are forced to move onshore by associated
        Ekman transport, which is balanced by an upward sea level inclination towards the coast,
        leading to westward currents along the coast (Holland, 2001).

        Maud Rise (MR), a massive seamount of ~200 km diameter with steep flanks that reach ~1200
m depth from the almost flat sea bottom (~5000 m), has a significant topographical influence
        on the Lazarev Sea water circulation. The MR center is located at ~65°S and ~2°30'E. Its steep
        elevation, as well as the influence of the WG, modify the vertical distribution of WDW through
        the water column (Carmack, 1974). Additionally, eddies produced by the WG create a warm
        area at the west flank of the MR that enhances the production of sensible heat, thereby
increasing vertical mixing (Holland, 2001; Leach et al., 2011). The pycnocline has been
        described as an energy reservoir or 'thermal barrier' (Martinson, 1990). The thermal barrier
        controls the heat fluxes from the depth and consequently the sea ice thickness (McPhee, 2003;
        Martinson, 1990). We can characterize these fluxes, by quantifying the amount of energy
        available through diapycnal and isopycnal diffusivity. Diapycnal mixing promotes a vertical
exchange of energy. Isopycnal mixing is induced by lateral circulation and dominated by
        eddies, mostly created by baroclinic instabilities (Leach et al., 2011). The heat fluxes and
        mixing levels may be affected by the interannual (typical period is four years) variability of the
        Antarctic Circumpolar Circulation in response to the meridional wind stress anomalies
        (Fahrbach and Beckmann, 2001). The combination of these factors results in conditions
favorable for the formation of an open-ocean polynya.

        An open-ocean polynya is an ice-free area surrounded by the winter sea ice pack. Such an
        occurrence is generally associated with a vertical oceanic circulation pattern in which
        upwelling warmer waters promote the melting process and prevent ice formation (Fahrbach
and Beckmann, 2001), thereby maintaining the ice-free region. A strong ocean-to-atmosphere
        heat flux also characterizes open-ocean polynyas, and they can have a strong regional
        influence. WSDW cooled and freshened significantly after the 1970s Maud Rise Polynya



(Gordon, 1981); water properties from the central Weddell Sea required over a decade to recover from the convection that occurred during those polynya years (McPhee, 2003).


Several processes have been proposed as drivers of this deep convection. Therefore, three main processes have been identified as potentially important as preconditioner and to sustain the convection in the Lazarev Sea. 1) An accumulation of WDW to provide a heat reservoir at depth (Lindsay et al., 2008; Martin et al., 2013; Cheon et al., 2015; Dufour et al., 2017; Kurtakoti et al., 2018); 2) a strong negative wind stress curl over the Lazarev Sea, which enhances the cyclonic WG and leads to warmer water upwelling (Cheon et al., 2015); and, 3) a destabilization of the upper ocean which promotes the exchange of heat and salt between different water layers, denominated as a thermobaric convection processes (Garwood et al., 1994). Thermobaric convection is a phenomenon related to the pressure dependence of the physical properties through the water column. With a weak pycnocline, a vertical displacement of cold water downward (or warm water upward) can lead to an unstable state in which the displaced parcel is denser (or lighter) than its surroundings. This effect can bring warm and salty water from deeper layers to the surface, without the necessity of a surface buoyancy flux.

Recent studies have identified an increase in surface salinity as key to providing an increase in the upward flux of warm water (Gordon 2014; de Lavergne et al., 2014; Kurtakoti et al., 2018). Dufour et al., (2017) pointed out that eddies and dense water overflows off the continental shelf play an essential role in Weddell Sea stratification, especially in the Lazarev Sea. Using high-resolution simulations, Kurtakoti et al., (2018) showed that the steep bathymetry at Maud Rise is key to developing the preconditioning for the occurrence of Maud Rise Polynyas, the precursors of Weddell Sea Polynyas. Ocean-ice-atmosphere interactions can lead to increased water density, a necessary condition to drive the overturning circulation (Bersch et al., 1992). Although there are many theories, the literature still has a lack of information to a complete description of the polynya formation and maintenance processes (Hirabara et al., 2012). An improved understanding of the polynya mechanisms is crucial due to the influence of polynyas on the global overturning circulation.

This study characterizes the dynamics of the Maud Rise Polynya by using, for the first time, *in situ* data recorded within the 2017 polynya event and supported by reanalysis results. This paper is organized as follows: Data used in this study (*in situ* and reanalysis) are described in Section 2. In Section 3 we describe our methodology. In Section 4 we discuss the results and the implication of the mixing levels necessary to produce the 2017 polynya including a discussion on the internal ocean structure, the horizontal and vertical level of mixing, and the heat fluxes. Finally, in Section 5 we draw some remarkable conclusions.


## 2. Datasets

For this study, we use *in situ* data recorded from January 2015 to January 2019 together with reanalysis data. We consider the 2017 Polynya event starting from September 14th 2017, when

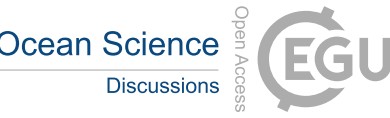

the sea ice concentration was the lowest, until December 1$^{st}$ 2017, when it was fully opened to
       the Southern Ocean.

## 2.1 Observational Data (in situ)

Observations south of ~60°S are sparse and biased towards summertime conditions. One of
       the initiatives to overcome this lack of data is the Southern Ocean Carbon and Climate
       Observations and Modeling (SOCCOM) project, administrated by the Princeton
       Environmental Institute and supported by the National Science Foundation through the
       Division of Polar and NOAA programs. The SOCCOM project deploys free-drifting profiling

floats to measure water properties such as temperature, salinity, and oxygen of the upper 1700
       m of the ocean. The measured data is transmitted via satellite each time when the float reaches
       the surface. If the surface is covered by ice, to prevent getting trapped, the float avoids the
       surface, stores the data until the next time it will reach the surface and descends for another
       round of profiling. When the float can eventually surface, these data are finally transmitted, it

is without tracking (position) information since the float relies on GPS. Over 200 SOCCOM
       floats have been deployed and are currently measuring the Southern Ocean
       (http://www.argo.ucsd.eduv, http://argo.jcommops.org). Four SOCCOM floats drifted into the
       Lazarev Sea (Fig. 1), and 2 of these drifted inside the 2017 Polynya area (ARGO ID 5904468
       and 5904471).


       The oceanographic temporal data series were collected over the region 59-67°S and 10°W-8°E
       (Fig. 1). The floats operate from the surface down to 1700 m, with a higher frequency sampling
       over the first 1000 meters. We considered two regions: one over Maud Rise Region (MRR)
       with the SOCCOM floats 5904468 (hereafter *68) and 5904471 (*71) and another at North

Maud Rise (NMR) with floats 5904467 (*67) and 5904397 (*97). The floats are equipped with
       a Conductivity Temperature Depth (CTD) unit. Also, chemical sensors measure dissolved
       oxygen, pH, nitrate, chlorophyll fluorescence and particulate backscatter. The manufacturer-
       stated accuracies for Salinity ($S$), Temperature ($T$), and pressure ($P$) are 0.01, 0.002 °C, and 2.5
       dbar, respectively (Riser et al., 2018). The $T$, $S$, and $P$ were used to calculate potential density

anomaly ($\sigma_\Theta$), Turner angle ($T_U$) and Buoyancy frequency ($N$), among others. The under-ice
       profiles without recorded latitude and longitude were estimated using linear interpolation. In
       this way, we capture the mean trajectory and the associated water properties, but neglect small
       eddies and topographical effects during those short periods.

We selected a representative SOCCOM float for each study area (*68 for the MRR, and the
       *97 for the NMR) to quantify the spatial properties of the polynya and the water column
       variability over the two different regions. We considered the data recorded from January 2015
       to January 2019 to track changes in the ocean dynamics before (preconditioning) and after
       (post-event) the appearance of the Polynya.


## 2.2 Model data output (HYCOM)


Velocity fields (*u* and *v*), necessary to calculate lateral mixing (described in section 3.1), were obtained from HYbrid Coordinate Ocean Model (HYCOM, www.hycom.org) model output
(Bleck, 2002) available at the HYCOM Data Servers (see https://hycom.org/dataserver). The global ocean model HYCOM is a primitive equation, which considers a general circulation model with high horizontal resolution and adaptive vertical coordinate system. This system uses a terrain-following coordinate at shallow water and z-level coordinates in the mixed layers and/or unstratified water (Bleck, 2002). The model uses the layered continuity equation to
make the transition from z-coordinates to terrain-following in a dynamically smooth way. This approach allows higher resolution near to the surface, as well in shallow regions. The horizontal resolution is 1/12° at the equator. To properly account for the *momentum*, heat and salt fluxes at the interface between ice and ocean, HYCOM has a coupled system with the Polar Ice Prediction System (Posey et al., 2008a) through the Earth System Modeling Framework
(ESMF, Hill et al., 2004). HYCOM also uses NCODA (Navy Coupled Ocean Data Assimilation) system to assimilate surface observations from satellites, including altimeter and Multi-Channel Sea Surface Temperature (MCSST) data, sea ice concentration and *in situ* profile data from XBTs, ARGO floats and moored buoys (Cummings, 2006; Cummings and Smedstad, 2013). As surface forcing, HYCOM uses the atmospheric forecast from NAVy
Global Environmental Model (NAVGEM, Hogan et al., 2014).

As HYCOM uses NCODA to assimilate the profiles from ARGO floats and propagate the information through the grid cell, we expected the output profiles from the model to possess high correlation with the *in situ* profiles. Model/data correlation was tested using T and S fields. In general, the correlation was higher than 0.7, with more than 60% of the profiles showing
correlation higher than 0.9. Since we obtained values higher than the threshold (0.7) for every profile used, we assumed the relationship between model and *in situ* profiles was sufficiently high to proceed with the use of velocity fields to analyze the internal ocean mixing.

**3. Ocean mixing methodology**


To assess the characteristics of the system, we estimate the Rossby number ($R_O$), which relates the ratio of inertial to Coriolis forces of the fluid:

$$R_O = \frac{U}{fL} \tag{1}$$

Where $U$ is the velocity scale, $f$ is the Coriolis parameter, and $L$ is the horizontal length scale.
We proceed to estimate the diapycnal diffusivity ($k_\rho$), a process that plays an essential role in controlling the vertical pattern of the ocean circulation, which ultimately determines the variability in energy between isopycnals. Mixing leads to an unbalanced pressure field that eventually results in the collapse and dispersion of the mixing waters through isopycnals (Thorpe, 2005). $k_\rho$ is estimated through the dissipation rate of turbulent kinetic energy (ε),
which is the rate of energy transferred from mechanical to caloric due to $N$ (buoyancy frequency) and the Thorpe scale ($L_T$) (Dillon, 1982).

$$\varepsilon = 0.64 L_T^2 \langle N \rangle^3 \tag{2}$$

$$L_T = \langle \partial_T^2 \rangle^{\frac{1}{2}} \; ; \; \partial_T = (Z - Z_s) \tag{3}$$




where $<N>_i$ is the mean value of $N$ in the reordered region of the overturn, $\partial_T$ is the Thorpe displacement, $Z$ is the depth, and $Z_s$ is the gravitationally stable depth where a fluid parcel originates. We applied the Galbraith and Kelley (1996) method to identify any overturning regions and to reject noisy data by reducing spiking. To validate the overturning, we calculate

the Root-Mean-Square (RMS) value between a linear sorting potential density and the temperature and salinity. Overturns where the maximum of RMS exceeded 0.5 were rejected following Galbraith and Kelley (1996). We apply the dissipation rate to estimate the vertical mixing rates, following the Osborn (1980) relationship:

$$k_\rho = \Gamma\varepsilon/N^2 \qquad (4)$$

where $\Gamma$ is the empirically defined mixing efficiency. Another process that modifies and controls the distribution of water properties is the isopycnal diffusivity ($k_h$), which quantifies the energy between isopycnals. A convenient tracer to observe these lateral anomalies is the

salinity ($S$), and the salinity anomaly ($S'$) on isopycnal surfaces (Munk, 1981). The isopycnal diffusivity is quantified as follows:

$$k_h = C_0 \varphi U_{rms} \qquad (5)$$

$$\varphi = \langle S'.S' \rangle^{\frac{1}{2}} / \langle |\nabla\{S\}| \rangle \qquad (6)$$

where $C_0$ is the mixing efficiency, $\varphi$ is the mixing length, $S' = S - \{S\}$, braces { } indicate a one year average in our study area, brackets ⟨ ⟩ a temporal average over full time period, and $U_{rms} = \langle (u - \{u\})^2 + (v - \{v\})^2 \rangle^{\frac{1}{2}}$ (Naveira Garabato et al., 2011). We can calculate

isopycnal lateral mixing from these equations, as described by Cole et al., (2015).

The surface of the Lazarev Sea is characterized by a cold and fresh surface layer (ASW) overlying the relatively warm and salty WDW, which makes the water column prone to developing a diffusive layering (Turner, 1973). The strength of the diffusive layering can be

characterized by the density ratio ($R_\rho$) and Turner Angle ($T_U$), defined as:

$$R_\rho = \beta\left(\frac{\partial S}{\partial Z}\right)\bigg/\alpha\left(\frac{\partial T}{\partial Z}\right) \qquad (7)$$

$$T_U = arctan\left(\frac{1+R_\rho}{1-R_\rho}\right) \qquad (8)$$


where $\alpha = \frac{1}{\rho}\left(\frac{\partial\rho}{\partial T}\right)$ and $\beta = \frac{1}{\rho}\left(\frac{\partial\rho}{\partial S}\right)$ denote the thermal expansion and saline contraction coefficients, respectively. $R_\rho$ estimates the contribution of thermal and salinity conditions to the water column stability (Tippins and Tomczak, 2003). A range of $0 < R_\rho < 1$ is related to diffusive convection processes, and $R_\rho>1$ is associated with salt fingering. Under this last




condition the turbulent motions may have been insufficient to disrupt the structure of the diffusive cores located in the area (Worster, 2004). Ruddick (1985) defines $T_U$ as a relation between temperature and salinity, to identify the level of stability in the water column ($-45° \leq T_U \leq 45°$ for a stable water column). These parameters are needed to calculate the heat flux ($F_H$) across the boundary layer. We calculated $F_H$ across the interface between the cold and

fresh surface layer and the warm and salty WDW following:

$$F_H = \lambda_{(\tau)} \left[ 1 - \tau^{\frac{1}{2}} R_\rho / 2 \left( 1 - \tau^{\frac{1}{2}} \right) \right]^{\frac{4}{3}} k \left( \frac{\alpha g}{\kappa v} \right)^{\frac{1}{3}} \Delta T^{\frac{4}{3}} \tag{9}$$

where $\lambda_{(\tau)}$ is an empirical function of $(\tau)$. $\tau$ represents the diffusivity ratio between salt and

heat, $k$ is the thermal conductivity of the fluid, $g$ is the gravitational acceleration, $\kappa$ is the thermal diffusivity, $v$ is the kinematic viscosity and $\Delta T$ the temperature difference between the layers (Worster, 2004). The consideration of $R_\rho$ and $F_H$ allows us to identify diffusion-convection processes within the study area.

## 4. Results and discussion

### 4.1 Internal structure

The upper zone of the Lazarev Sea system (< 1000 m depth) can be described as a two-layer

system, each layer weakly stratified and separated by a sharp interface located at ~80 – 140 m depth. The shallow layer (called ASW, ~ <120 m) is cold and fresh above the relatively warm and salty subsurface WDW layer (Fig. 2c-f). We identify a remarkable change of conditions between adjacent profiles confirming diffusive processes. The data recorded from Dec. 2014 to Dec. 2018 show an ephemeral, shallow pattern during summer, which has higher

temperatures (max. of 2.93 °C) and lowest levels of salinity (min. of 33.67) at ~20 m depth. The summer seasonal layer reaches from the surface down to ~ 70 m depth. The high temperature and lower salinity observed in this shallow layer during summer 2017 are uncommon compared to the same season of previous and following years. The seasonal and interannual ocean properties seem to persist over the observed time scale in both regions.


During wintertime, the surface waters over MRR have a higher potential density anomaly ($\sigma_0$ = 27.79 kg m$^{-3}$) compared to NMR ($\sigma_0$ = 27.55 kg m$^{-3}$). These shallow layer values could be a result of sea ice melting that helps to maintain the sharp pycnocline (Fig. 2g). Over the MRR, in June 2017, the surface water ($T$ = -1.8 °C) was close to the freezing point ($T_f$ = -1.9 °C).

During this winter period, the shallow water layer extends to ~115 m depth with a temperature of -1.74°C and salinity of 34.49. From this point on, the physical properties increase rapidly until the depth of 300 m, reaching 0.99 °C and 34.72 ($\sigma_0$ = 27.83 kg m$^{-3}$). At this depth, the temperature smoothly decreases again reaching values of 0.47 °C and 34.70 at 900 m. At the same depth 100 km distant, a maximum value of 0.57 °C was recorded in September 2017,

when the 2017 Polynya event was reported.

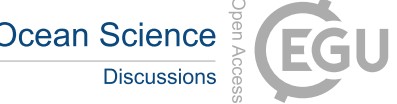

Before the 2017 Polynya event, the sub-surface water temperature increases from -1.62 ºC to -0.44 ºC at 90 m depth, producing a potential instability in shallow areas. From looking at the stratification, $N$, a constant layer of high stratification between 90 and 140 m depth is detected at the NMR (Fig. 2i), a layer that became weak (or broken) during the winter periods over the MRR. As the summer season advances and winter approaches, the depth of stability of this layer decreases, creating conditions that enable vertical processes; the internal structure has a weak stratification below the thermohaline (> 200 m). The Turner angle varied through the study period, but there was a well-defined layer of static stability between 90 and 170 m depth. We highlight the values of $T_U < -45°$ (Fig. 2k-l), which show regions where the water column is unstable to diffusive convection. There is a small variability in potential temperature of the water masses between consecutive profiles, an effect produced by diffusivity and convection processes.

The conditions of weak stratification, upwelling, and small Rossby number relative to the MR length scale ($R_o = 0.02$ weak inflow) are conducive to the formation of a Taylor cap (Ou, 1991; Alverson and Owens, 1996). The trapped water can produce an anomalous increase in density in the intermediate water over MR. During the 2017 Polynya event, the density at 120 m is higher ($\rho_{2017} = 27.84$ kgm$^{-3}$) compared with the previous years ($\rho_{2015} = 27.78$ kgm$^{-3}$) at the same depth. This density variation induces deep convection and overturning circulation, both critical factors for Polynya maintenance. Once the mixing layer is deep enough, Taylor cap movements can increase the convection process (Meredith et al., 2015).

The months prior to and during the Polynya event showed shallow surface water salty, cold, and dense; such significant changes are associated with convection processes that provide preconditioning for the Polynya event (Kurtakoti et al., 2018). Colder and denser water sinks, allowing the warmer water to rise and take its place, which creates the conditions that are favorable for the polynya formation. The salinity increment could be generated by the brine rejection from the first stage of sea ice formation, which destabilizes the upper ocean as is reflected in the values of $N$ (Fig. 2i). The small rate of increase in temperature along isopycnals suggests that diapycnal and isopycnal mixing is occurring. These observations demonstrate how the system is predisposed to be affected by vertical convection processes that exchange properties between shallow and sub-surface waters.

As the summertime progresses, the transference of heat and salinity properties from ASW to the sub-surface thermal barrier occurs, which works as an energy reservoir. This exchange happens at the end of the summer when the surface layer reaches the same depth as the sharp boundary, which brings the transfer of energy to an abrupt halt. Temperature variations could be the trigger for such lateral and vertical mixing, increasing the heat ventilation as identified by the temperature variability (up to 1°C) at the same depth from autumn to wintertime (Fig. 2c).

**4.2 Mixing processes**





In this section, we evaluate the diapycnal (vertical) and isopycnal (lateral) mixing that occurred during the 2017 Polynya event.

### 4.2.1 Diapycnal diffusivities

Diapycnal diffusivity calculated from equations (4) and (5) is presented in Figure 3. Over the NMR, a visible permanent layer of low diffusivity levels fluctuates from ~ 50 m to ~ 160 m according to the season (Fig. 3b). The full period average value of the diapycnal diffusivity is $k_\rho \approx 3 \times 10^{-4}$ m$^2$s$^{-1}$. The sub-surface WDW and the summer layer contain the highest diffusivity rates at around ~50 m ($k_\rho \approx 5 \times 10^{-2}$ m$^2$s$^{-1}$,), and ~250 m depth ($k_\rho \approx 3 \times 10^{-2}$ m$^2$s$^{-1}$),
respectively. There is no well-defined permanent low diffusivity layer over the MRR (Fig. 3a). Instead, some patches increase with depth from ~30 m to ~110 m as the summer season advances but stop when winter arrives. At the MRR the highest diapycnal diffusivity calculated over the 2017 Polynya-growing period is $k_\rho \approx 1 \times 10^{-2}$ m$^2$s$^{-1}$; this value does not change significantly over the full water column, due the lack of sea ice cover that protect the area from
wind that produce waves in the area.

Mixing along isopycnal surfaces has a continuous relation with thermobaric convection, considered here through the thermobaric coefficient (T$_{\Theta b}$). As the vertical properties change, the thermal and density coefficients also change. The highest thermobaric coefficients located
in shallow waters (Fig. 3c, d) are related to low thermal expansion and high salinity contraction (Garwood et al., 1994). Thermobaric processes enhance the descending plumes of properties that increase the water density; these density changes are reflected as a compressed isopycnal observed during summer time (Fig. 2g). The density variability decompensates intermediate waters producing periods of lighter waters over MRR. The gradual reduction in potential
temperature variance on isopycnals is indicative of mixing processes, due to the water masses convergence. This convergence of water with the same density but different temperature and salinity produces mixing via convection processes.

There are no other measurements of $k_\rho$ during a Polynya event to compare with, but the low
mixing layer values (Fig. 3b) are similar to those recorded by Naveira Garabato et al., (2004a) in the Southern Ocean (~ 3 x 10$^{-4}$ - 1 x 10$^{-2}$ m$^2$s$^{-1}$), and Cisewski et al., (2008, 7 x 10$^{-4}$ m$^2$s$^{-1}$) in the upper layers of the ACC. During the Polynya event, float *68 was located on the west flank of MR. The sensible heat production in this region is intensified by an accumulation of warm water when compared to the east side (Fig. 2c), responsible of the increase of the vertical
mixing (Holland, 2001; Leach et al., 2011). Another component of mixing (lateral), was evaluated through the velocity fields, which can also change under the influence of the MR. The velocity fields and associated lateral mixing will be described in the next section.

### 4.2.2 Isopycnal diffusivities

We highlight the velocity field layers at the depths of 50, 150, and 250 m to illustrate the general behavior of the water masses previously described. An example over the Lazarev Sea during November 2017 is presented in Fig. 4c. The fastest currents occurred in the shallow waters (~





50 m), between values $U=$ 0.56 and $U=$ 0.04 cms$^{-1}$ westward. At the NMR, these currents
decrease with depth, reaching values of $U=$ 0.42 – 0.02 cms$^{-1}$ close to the thermal barrier (Fig.
4d). The velocity field during the Polynya event is mainly westerly (Fig. 4c).

A constant front that interacts with the MR from the east is visible in the velocity fields. The
steep slope of the MR east flank squeezes the isopycnals, inducing upwelling of sub-surface
waters. This upwelling is responsible for elevating the temperature of the shallow surface
waters slightly. Topographic interaction on the east flank also induces anticyclonic eddies with
upward domed isopycnals that enable heat and salt fluxes. Intensification of eddies affects the
water residence time; if it is high, the system upwells bringing more heat and salt from sub-
surface waters to the surface. On the other hand, cyclonic events may produce ice divergence
(Holland, 2001).

Considering the instability in the water column, we used the velocity fields to quantify the
isopycnal variability in the water column (Fig. 4c-e). As expected, this is a continuous process
with higher lateral mixing values in shallow waters regarding the effects of the Maud Rise.
Mixing values decreases at deeper layers. The average value for the MRR at 50 m is $k_h \approx 6$
m$^2$s$^{-1}$, and at 250 m is $k_h \approx 1$ m$^2$s$^{-1}$. There are some peaks of energy during wintertime,
specifically during the 2017 Polynya event, reaching maximum values of $k_h \approx 30$ m$^2$s$^{-1}$. As
there is a lack of direct measurements in this MR Polynya, we searched and compared the
values with other estimations that presented similar ocean conditions but in a different context.
The values we got agree with the findings of Leach et al., (2011). Ledwell et al., (1998),
quantified isopycnal diffusivities of $k_h \approx 5$ m$^2$s$^{-1}$, enough to dissipate  energy around ~ 1 – 10
km. The sampling spacing recorded by the floats was around $\pm$ 20 km. We can use the scale-
diffusivity dependency relationship to estimate the incidence of the horizontal eddy
diffusivities. Considering that lateral mixing is scale-dependent (e.g., Ledwell, 1998; Hibbert
et al., 2009), a large amount of energy can be expected to be distributed over the full MRR,
and this energy is then available to supply the thermal barrier. The range of values calculated
during the Polynya event, 30 – 150 m$^2$s$^{-1}$, are in accordance with those expected in the presence
of mesoscale eddies. The presence of mesoscale eddies in the MRR and the appearance of the
low stability values related to the weakly stratified water column during wintertime might
create the preconditioning for the Polynya event.

Lateral diffusivity can be related to the potential density anomaly recorded over the water
column. Regarding the density profilers, we can relate those values with different water layers
over the study period (Fig. 5). During 2016, the energy over the full MRR reaches a maximum
of $k_h \approx 70$ m$^2$s$^{-1}$, and over NMR a maximum of $k_h \approx 105$ m$^2$s$^{-1}$. During 2017, the energy
distribution over different densities decreases, with a remarkable peak during the polynya
months (Fig. 5c). Throughout this event, the lateral energy calculated from float *68 (MRR)
reaches a maximum of $k_h \approx 150$ m$^2$s$^{-1}$, in a density range of 27.80 – 27.90 kgm$^{-3}$, and from float
*97 (NMR), a maximum of $k_h \approx 90$ m$^2$s$^{-1}$, in a range of 27.70 – 27.80 kgm$^{-3}$. These values
correspond to the WDW and the location of the thermal barrier, respectively. Leach et al.,
(2011) described similar variations with estimated values of $k_h \approx 140$ m$^2$s$^{-1}$, in the density range
of 27.77 – 27.82 kgm$^{-3}$. Now that we identify the relevance of the energy reservoir for the ocean

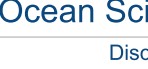
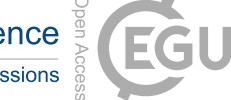

preconditioning which is a prerequisite to develop a polynya event, we proceed by describing the heat fluxes associated with the thermal barrier.


### 4.2.3 Strength of the Heat flux

The sudden interruption of the summer layer at the middle of March of each year could be produced by the full transfer of energy from the summer layer to the permanent intermediate

layer through the sharp transition of the thermal barrier. This energy reservoir determines the magnitude of $F_H$. Heat transfer from the ocean to the atmosphere when the Polynya was fully developed (size ~ 300 x $10^3$ km$^2$) varies from ~50 – 220 Wm$^{-2}$; values that support the deep-water convection produced in the area, in agreement with Gordon, (1982) and Moore et al., (2002).


Over the full study region, the sub-surface water induces convection processes that modify the distribution of temperature and salinity through sinking plumes processes, such as brine salt rejection , which influences deeper waters (Fig. 2). This process occurs more over the NMR, while over the MRR diffusivity increases with depth (~ 750 m depth) during the Polynya event

(Fig. S1). To characterize the effect of the thermal barrier on the surrounding waters, we calculated the heat flux ($F_H$). Overturning processes can be illustrated by $L_T$ and $R_\rho$ (Fig. 6). There is an outstanding variability located around 130 m depth, showing a clear difference between surface and intermediary waters where occurred the heat exchange.

$F_H$ was calculated over the Lazarev sea system (Fig. 6). The constant heat flux into the lower boundary, from sub-surface to shallow waters, is characteristic of a diffusive regime forced by convection processes. Once the convection is produced, the vertical mixing brings the energy reservoir to shallow waters, producing deep ventilation through this unstable system. The thermal barrier can be identified throughout the year, fluctuating between ~ 50 to ~ 150 m

depth. We observed an opposite heat flux layer near the surface that increases its depth, starting at ~10 m in summer reaching ~70 m in spring. The summer regime produces an inverted condition of heat flux, transferring energy from the shallow seasonal layer to the sub-surface waters. The transfer of energy by the summer layer suddenly stops each year in the middle of March when the two heat fluxes boundaries meet at around ~ 100 m depth. The $F_H$ gradient is

higher over the MRR compared with the NMR (Table 1), reflecting the important role of the MR. The energy is accumulated during the full year, exhibiting the highest values at the end of summer and the lowest in winter. The energy concentrations and variability at the beginning of wintertime exhibit the highest rates of heat ventilation. The calculated values of $F_H$ agree with the literature (e.g., Muench et al., 2001).


### 5. Conclusions

The internal structure of the ocean layer in the Maud Rise polynya area shows an almost uniform distribution of properties throughout the year, with changes in the sub-surface waters

occurring before the Polynya formation. In the Lazarev Sea, SOCCOM floats *68 and *97





recorded similar patterns of density, temperature, and salinity distributions over the water column, with the main differences in maximum and minimum values. In wintertime, the potential density anomaly was around $\sigma_{0*68} = 27.87$ kg m$^{-3}$ at 200 m depth, with a maximum peak of $\sigma_{0*97} = 27.81$ kg m$^{-3}$ at the same depth in June 2017, prior to the Polynya event. Under

this density condition, the temperature was low but above the freezing point (-1.9 $^{o}$C). Besides, the salinity flux from summertime sea ice melting dominates, setting ocean preconditioning for the Polynya development. Comparing the density with the temperature and salinity, the latter drives the changes in density leading to a weak stratification. This condition promotes convection, as reflected by the accumulated instability in the sub-surface waters.


Among the relevant oceanic factors, we highlight three that contribute to the open-ocean Polynya preconditioning: 1) A higher level of instability due to the lateral and vertical transports of heat and salt (Section 4.3). 2) Pycnocline fluctuations are affected by lateral and vertical mixing, which increases the bottom-up fluxes of heat and salt (Fig. 2). These patterns

detected in the temperature and salinity observations confirm that a balance between the lateral and vertical mixing level occurs over the shallow waters (~<300 m depth), maintained by convection processes. And 3) The influence of the MR is undeniable; over MR we observed a decrease of the depth at which the thermal barrier is located by almost ~40 m (~120 m at the NMR, and ~80 m at the MRR).


Thermobaric convection produces the dissipation of energy through almost the full water column where records are available (1700 m). The tendency for thermobaric convection is higher during the Polynya event compared to previous years, where such instability is a recurrent process during wintertime. Another finding relates to the modification of the thermal

barrier over the MRR. The weak stratification of the thermal barrier is an essential ingredient for the Polynya preconditioning, where vertical mixing initiates the dissipation of energy through the water column. This process influences the internal structure and affects surface layers in the region where the Polynya develops. The final aspect is the production of fresh shallow water resulting from the melting of sea ice that can produce denser deep waters due to

the sinking plumes. This overlapping of water masses produces extensive overturning. Together this dataset represents a thorough oceanic description of polynya formation and maintenance.

Ocean forcing alone cannot trigger polynya features; appropriate atmospheric conditions are

needed to promote the development of such a structure. Recent studies have highlighted the need for a negative wind stress curl, which can derive cyclonic thermal winds in the lower layer of the atmosphere to produce ice divergence (e.g. Cheon et al., 2015; Kurtakoti et al., 2018; and Francis et al., 2019b). Francis et al., (2019b) have identified the particular role of atmospheric cyclones in triggering a polynya event by creating ice divergence and ocean

upwelling increasing both the size and the period of the polynya. When the atmospheric and oceanic conditions overlap, a polynya event can be expected. Often these conditions occur at different times, producing a low sea ice concentration in the region, but they are unable, separately, to produce and maintain a polynya event. This statement is supported by the





diapycnal and isopycnal analysis presented here, as well as by the findings by Hirabara et al.,
510  (2012).

**Acknowledgments**

This research was supported by the Center for global Sea Level Change (CSLC) of NYU Abu
Dhabi Research Institute (G1204) in the UAE. Data collected by *NSF's Southern Ocean
Carbon and Climate Observations and Modeling (SOCCOM) Project under the NSF Award
PLR-1425989, with additional support from NOAA and NASA. Logistical support for this
project in the Antarctic was provided by the U.S. National Science Foundation through the
U.S. Antarctic Program.* We thank all colleagues at the CSLC for critical review and support.

**Appendix**

Table A1. Acronym table. Relevant acronyms in order as was shown in the manuscript.

| Acronym | Definition |
| --- | --- |
| WS | Weddell Sea |
| WG | Weddell Gyre |
| ASW | Antarctic Surface Water |
| WDW | Warm Deep Water |
| ACC | Antarctic Circumpolar Current |
| WSDW | Weddell Sea Depth Water |
| WSBW | Weddell Sea Bottom Water |
| AABW | Antarctic Bottom Water |
| MR | Maud Rise |
| SOCCOM | Southern Ocean Carbon and Climate Observations and Modelling |
| MRR | Maud Rise Region |
| NMR | North Maud Rise |
| $\sigma_\Theta$ | Potential density anomaly |
| Tu | Turner angle |
| N | Buoyancy frequency |
| $\varepsilon$ | Dissipation rate |
| $L_T$ | Thorpe overturning scale |
| $\partial_T$ | Thorpe displacement |
| $k_\rho$ | Diapycnal diffusivity |
| $k_h$ | Isopycnal diffusivity |
| $\varphi$ | Mixing length |
| $R_\rho$ | Density ratio |
| $F_H$ | Heat flux |




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



**Tables**

Table 1. Heat fluxes values over the Lazarev Sea.

| Heat flux | | | | | | |
|---|---|---|---|---|---|---|
| SOCCOM | Max. | | | Min. | | |
| | Value (Wm$^{-2}$) | Depth (m) | Date | Value (Wm$^{-2}$) | Depth (m) | Date |
| MRR *68 | 0.295 | 125 | Aug. 23 2017 | -0.703 | 40 | Feb. 9, 2017 |
| NMR *97 | 0.075 | 95 | Jun. 26 2017 | -0.170 | 40 | Mar. 4 2017 |

**Figures**

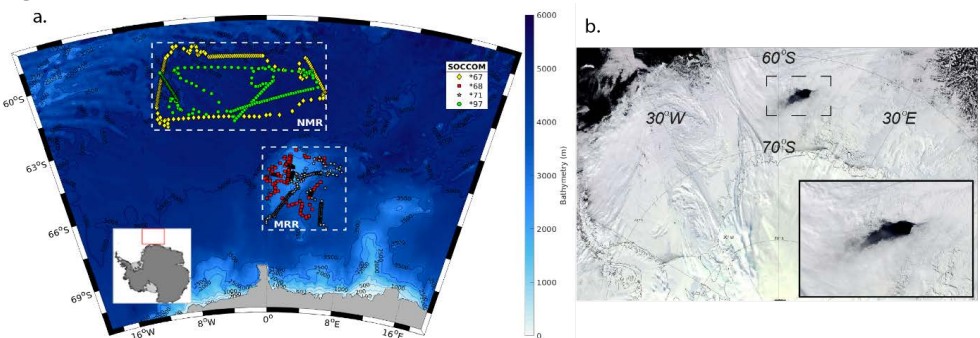

**Figure 1**. a. The Lazarev Sea area where the 2017 Maud Rise Polynya occurred. The colored
dots give the positions of the SOCCOM floats 5904397 (green circle), 590447**1**(gray star),
590446**8** (red square), and 590446**7** (yellow diamond). Location of the study area (inset). The
white dashed boxes correspond to the regions considered in this study, North Maud Rise
(NMR) and Maud Rise Region (MRR). b. MODIS Satellite image of the 2017 Polynya event
on September 23 2017. Inset. Zoom of the 2017 Polynya event. Images credit NASA
worldview.





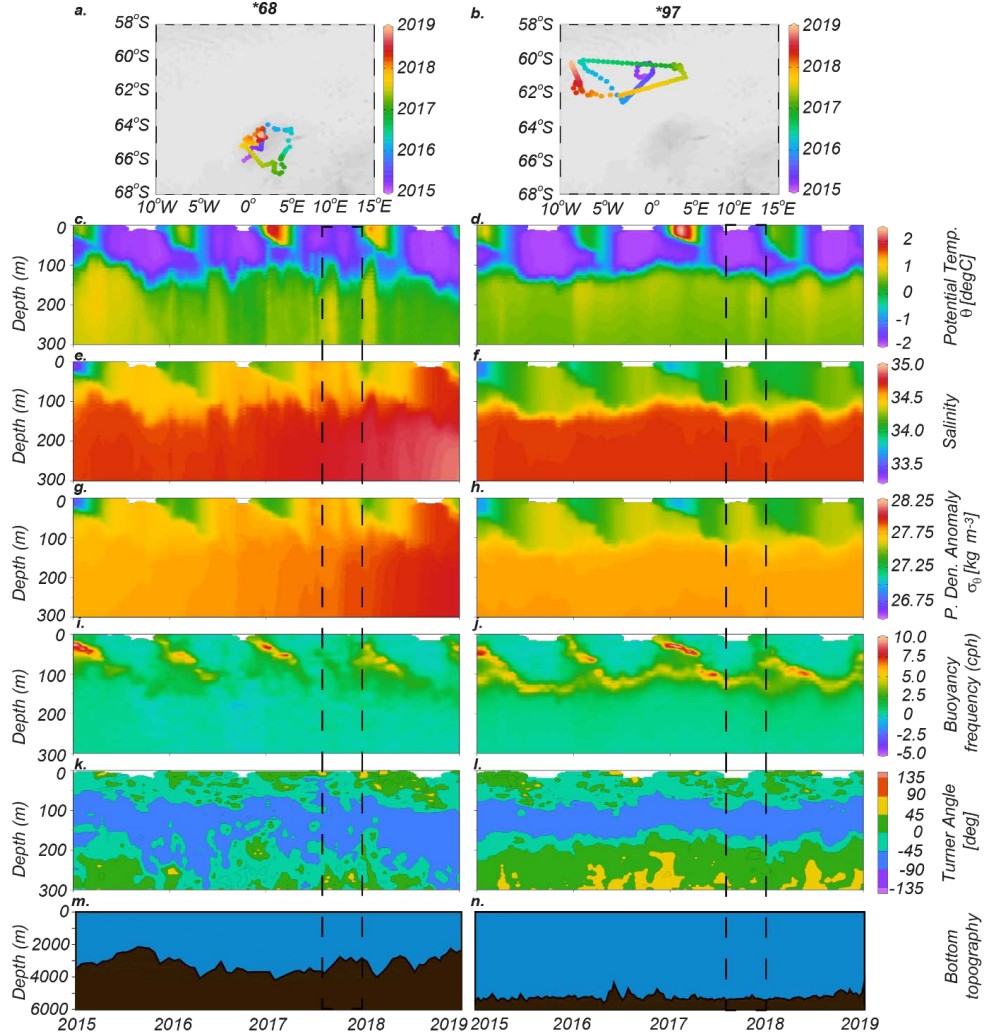

**Figure 2**. From top to bottom: Float location (a, b); potential temperature ($T$ in ºC) (c, d); salinity (S) (e,f); potential density anomaly ($\sigma_\Theta = \rho_\Theta - 1000 \, \text{kg m}^{-3}$) (g, h); buoyancy frequency ($N$ in cycles per hour) (i, j); Turner angle ($T_U$ in degrees) (k, l); bathymetry sections (meters) (m, n) for the data time series of SOCCOM *68 (MRR) and *97 (NMR), respectively. The data span January 2015 – January 2019 and profiles depths are shown from 0 to 300 m. Dashed lines delimit the period of the 2017 Polynya event. Profiles reaching 1000 m depth are shown in supplementary material Fig. S1.




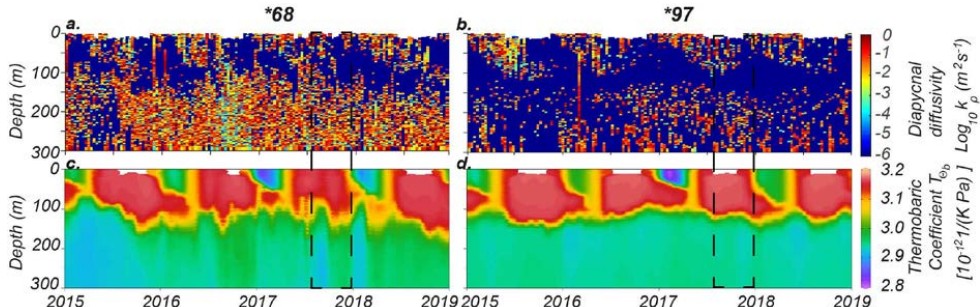

**Figure 3**. Diapycnal diffusivity ($k_\rho$ in m²s⁻¹, note the logarithmic scale, a, b), and thermobaric
coefficients ($T_{\Theta b}$ in [$10^{-12}$1/(KPa)], c, d) for SOCCOM *68 (MRR, a), and *97 (NMR, b). The
data span January 2015 – January 2019 and the profile depths are shown from 0 to 300 m.
Dashed lines delimit the period of the 2017 Polynya event.

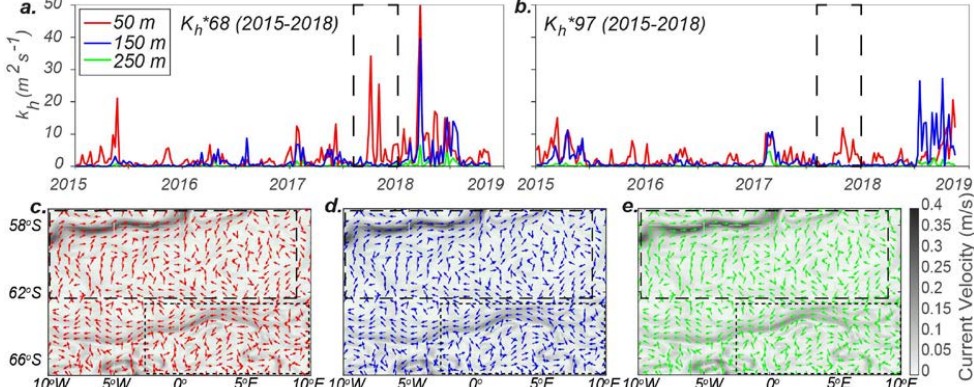

**Figure 4**. (a) Average isopycnal mixing level from January 2015 – January 2019 at different
depth layers (50, 150, and 250 m) for *68 and (b) *97. (c) Velocity fields over the Lazarev Sea
on Nov. 2017 at 50 m, (d) 150 and, (e) 250 m depth. The vertical dashed lines in (a) and (b)
mark the period of the 2017 Polynya event. Dashed and dotted squares delimit the influence
area of *97 and *68 respectively on c, d, and e.



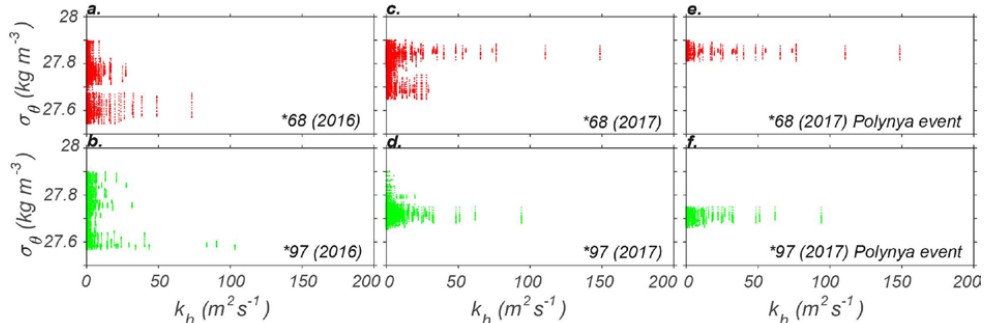

**Figure 5**. Lateral diffusivity vs. potential density anomaly, for SOCCOM float *68 (MRR), and *97 (NMR). (a, b) Average lateral energy during 2016, (c, d) 2017, and (e, f) 2017 Polynya months.

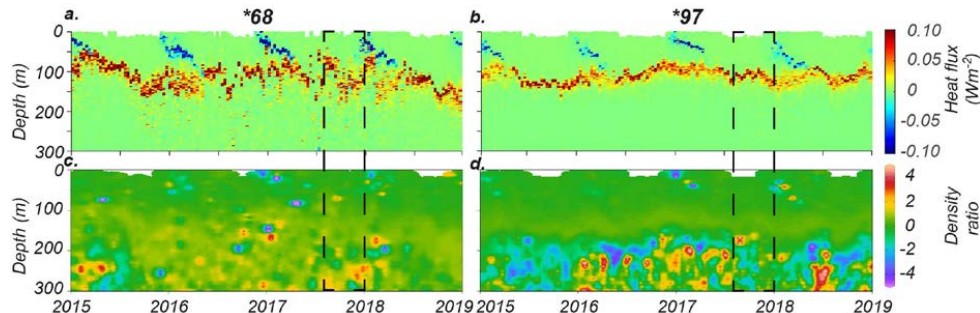

**Figure 6**. Time series of heat flux ($F_H$ in Wm$^{-2}$), and density ratio ($R_\rho$ in m), for the SOCCOM float data from *68 (a, c), and *97 (b, d) for the MRR and NMR, respectively. The data span January 2015 – January 2019 and the profiles are shown for depths from 0 to 300 m. Dashed lines mark the period of the 2017 Polynya event.