# Peer review of "Characterization of Ocean Mixing and Dynamics during the 2017 Maud Rise Polynya Event"

_Ocean Science, 2019_

## Referee Comment (RC1) · Anonymous Referee #1 · 12 Jul 2019

Dear editor,

I recommend that the manuscript be sent back for Major Revisions. The authors present and analyze unique observations of ocean properties during an open ocean polynya in the Lazarev Sea. However, the manuscript is not written very clearly and the results are presented in a confusing way. Moreover, the manuscript focuses exclusively on convection and various mixing processes but does not explore wind-driven advection, that may be constrained using reanalysis products. That is despite the fact that the title of the manuscript refers broadly to "dynamics" during the polynya. It seems that the authors do not present and test a hypothesis or a set of hypotheses about the causal chain of mechanisms that give rise to such open ocean polynyas.

Major issues:

[Figure]

1) The manuscript does not clearly present the observations and the analysis in the context of testing a hypothesis about the mechanisms behind a Polynya formation although it hints to such possible mechanisms in a very confusing manner. It seems that the authors are aware of ways in which their observations and analysis fit into the broader picture, but are not communicating this efficiently to the reader. I have not worked on the formation of open ocean polynyas, but my general understanding is that multiple processes with positive feedback mechanisms are at play, and this makes distinguishing cause and effect difficult. Could the authors' observations help disentangle the chain of events triggering and sustaining this open ocean polynya?

2) The manuscript title refers broadly to dynamics but the analysis focuses exclusively on convection and mixing processes. The contemporaneous anomalies in wind driven circulation are not given attention. Could the authors consider anomalies in the wind-driven circulation from reanalysis? Or alternatively, they could narrow down the scope of the paper, but be clear from the start that they are not fully exploring the dynamics of polynya formation.

3) The manuscript needs serious proofreading by the authors. This is not a minor issue because the text can be confusing at times. I may accept to review an updated version only if the quality of the text is substantially improved! That is why I indicated that I am not willing to review this again.

4) The introduction includes a broad overview but does not emphasize the important role of the halocline, the salt-stratification that allows a vertical temperature inversion (e.g., lines 59-65). Also, the introduction does not highlight differences between coastal and open ocean polynyas.

Minor issues:

Lines 16, 160, 163, 187, etc. You switch between present and past tense, but maybe you should stick to using present tense consistently throughout the text.

Line 95 and others. You vaguely talk about "physical properties" when you can be specific that you mean density.

Line 64, Line 368 and other instances – you talk about "production of sensible heat" when you mean "transport" and "release"

Line 100 "providing" –> "facilitating"

Line 114 "within" –> "during"

Line 119 You do not have to keep the reader waiting. Briefly state what we should expect.

Line 55 "by associated Ekman transport" – awkward phrasing

176 "near to the surface" –> "near the surface"

Line 200. Diapycnal "diffusivity" is not "a process." Diffusion is a process, while diffusivity is an inherent characteristic of the system.

Section 3 title. Why do you refer to the following as "methodology?" It seems that you are doing an overview of theory.

Lines 113, 137-140, 193-194, 202-206, 286, 289, 297, 325-326, 349-353, 394, 444, 450 – awkward or confusing phrasing

Line 226 "quanitified as" –> "defined as"

Line 271 expand the abbreviation ASW to explain what it stands for

Line 301 drop "a"

Line 335 – if the isopycnals are steep, then there is both a lateral and a vertical component to isopycnal mixing. So I would not label diapycnal and isopycnal mixing as vertical and lateral.

---

## Author Comment (AC1) · 17 Jul 2019

Dear referee #1, Thank you for reviewing our manuscript. We found the comments and suggestions useful, and we respond to them as indicated in our point-by-point answers below. The aim of this work is to describe and create a temporal mixing map in the area that allows us to illustrate the ocean characteristics prior to and during the formation of the Maud Rise Polynya. Therefore, we would like to change the manuscript title to "Characterization of Ocean Mixing during the 2017 Maud Rise Polynya event", eliminating the word dynamics. We present our hypothesis in lines 94-98. In the next version of our manuscript, we will highlight thermobaric convection, and clarify any confusion regarding polynya dynamics.

[Figure]

Major issues: 1) We are rewriting the confusioning parts of the manuscript that you mentioned, and are clarifying our hypothesis to present more clearly the role of thermobaric convection in driving energy exchange at the thermal barrier. This process is just one component of the ocean preconditioning that forms part of the polynya formation puzzle. As we discuss in lines 499 – 510, wind and atmospheric effects are needed to open the polynya. Moreover, Francis et al., 2019, have confirmed that atmospheric events trigger the opening of the polynya. Therefore, we focus here on the ocean part, using rare measurements to describe the ocean preconditioning that occurs in this region.

2) Yes, it was a mistake to choose 'dynamics' for the title. We will change the title by deleting the word 'dynamics'. By doing this, we are not narrowing down our scope, just clarifying our focus subject: an assessment of the ocean conditions that make Maud Rise susceptible to a Polynya opening, based on a new and rare dataset.

3) We will check the full manuscript to make sure that the present tense is used consistently and to clarify our ideas as best we can. But we take your suggestions into account in our revision.

4) Thanks for the comments. In our introduction, we emphasize the role of the pycnocline in relation to the thermal barrier, which is why we focus on the thermocline rather than the halocline. We will rewrite this part to mention the important role of the halocline, but will give more emphasis to the thermohaline.

Minor issues:

Lines 16, 160, 163, 187, etc. You switch between present and past tense, but maybe you should stick to using present tense consistently throughout the text.

OK, we will check all tenses throughout the manuscript to check for consistency and will stick to the present tense.

Line 95 and others. You vaguely talk about "physical properties" when you can be

specific that you mean density.

OK, we will check over the full manuscript and we will be more specific when we talk about physical properties.

Line 64, Line 368 and other instances – you talk about "production of sensible heat" when you mean "transport" and "release"

OK, we will change it.

Line 100 "providing" –> "facilitating"

OK, we will change it.

Line 114 "within" –> "during"

OK, we will change it.

Line 119 You do not have to keep the reader waiting. Briefly state what we should expect.

OK, we will add a sentence to conclude that thermobaric convection in an important driver of stability and exchange of fluxes in the thermal barrier.

Line 55 "by associated Ekman transport" – awkward phrasing

OK, we will change it.

176 "near to the surface" –> "near the surface"

OK, we will change it.

Line 200. Diapycnal "diffusivity" is not "a process." Diffusion is a process, while diffusivity is an inherent characteristic of the system.

OK, we will change it.

Section 3 title. Why do you refer to the following as "methodology?" It seems that you

are doing an overview of theory.

This is the theory and steps we follow to quantify the variables in this work.

Lines 113, 137-140, 193-194, 202-206, 286, 289, 297, 325-326, 349-353, 394, 444, 450 – awkward or confusing phrasing

OK, we will rewrite this part.

Line 226 "quanitified as" –> "defined as"

OK, we will change it.

Line 271 expand the abbreviation ASW to explain what it stands for

OK, we will expand it.

Line 301 drop "a"

OK, we will change it.

Line 335 – if the isopycnals are steep, then there is both a lateral and a vertical component to isopycnal mixing. So I would not label diapycnal and isopycnal mixing as vertical and lateral.

The isopycnals are not steep. The isopycnals squeeze because of the steep slope of the Maud Rise in an area approx. ∼100 km, to keep talking about isopycnal and diapycnal mixing.

Please also note the supplement to this comment:
https://www.ocean-sci-discuss.net/os-2019-41/os-2019-41-AC1-supplement.pdf

―――――――――――――――――――

**Supplement:**

Author answers to anonymous referee #1 are inserted in blue.

**Anonymous Referee #1**

Dear editor,

I recommend that the manuscript be sent back for Major Revisions. The authors present and analyze unique observations of ocean properties during an open ocean polynya in the Lazarev Sea. However, the manuscript is not written very clearly and the results are presented in a confusing way. Moreover, the manuscript focuses exclusively on convection and various mixing processes but does not explore wind-driven advection, that may be constrained using reanalysis products. That is despite the fact that the title of the manuscript refers broadly to "dynamics" during the polynya. It seems that the authors do not present and test a hypothesis or a set of hypotheses about the causal chain of mechanisms that give rise to such open ocean polynyas.

Dear referee #1,

Thank you for reviewing our manuscript. We found the comments and suggestions useful, and we respond to them as indicated in our point-by-point answers below.

The aim of this work is to describe and create a temporal mixing map in the area that allows us to illustrate the ocean characteristics prior to and during the formation of the Maud Rise Polynya. Therefore, we would like to change the manuscript title to "Characterization of Ocean Mixing during the 2017 Maud Rise Polynya event", eliminating the word dynamics.

We present our hypothesis in lines 94-98. In the next version of our manuscript, we will highlight thermobaric    convection, and clarify any confusion regarding polynya dynamics.

**Major issues:**

1) The manuscript does not clearly present the observations and the analysis in the context of testing a hypothesis about the mechanisms behind a Polynya formation although it hints to such possible mechanisms in a very confusing manner. It seems that the authors are aware of ways in which their observations and analysis fit into the broader picture, but are not communicating this efficiently to the reader. I have not worked on the formation of open ocean polynyas, but my general understanding is that multiple processes with positive feedback mechanisms are at play, and this makes distinguishing cause and effect difficult. Could the authors' observations help disentangle the chain of events triggering and sustaining this open ocean polynya?

We are rewriting the confusioning parts of the manuscript that you mentioned, and are clarifying our hypothesis to present more clearly the role of thermobaric convection in driving energy exchange at the thermal barrier. This process is just one component of the ocean preconditioning that forms part of the polynya formation puzzle. As we discuss in lines 499 – 510, wind and atmospheric effects are needed to open the polynya. Moreover, Francis et al., 2019, have confirmed that atmospheric events trigger the opening of the polynya. Therefore,  we focus here on the ocean part, using rare measurements to describe the ocean preconditioning that occurs in this region.

2) The manuscript title refers broadly to dynamics but the analysis focuses exclusively on convection and mixing processes. The contemporaneous anomalies in wind-driven circulation are not given attention. Could the authors consider anomalies in the wind-driven circulation from reanalysis? Or alternatively, they could narrow down the scope of the paper, but be clear from the start that they are not fully exploring the dynamics of polynya formation.

Yes, it was a mistake to choose 'dynamics' for the title. We will change the title by deleting the word 'dynamics'. By doing this, we are not narrowing down our scope, just clarifying our focus subject: an assessment of the ocean conditions that make Maud Rise susceptible to a Polynya opening, based on a new and rare dataset.

3) The manuscript needs serious proofreading by the authors. This is not a minor issue because the text can be confusing at times. I may accept to review an updated version only if the quality

of the text is substantially improved! That is why I indicated that I am not willing to review this again.

We will check the full manuscript to make sure that the present tense is used consistently and to clarify our ideas as best we can. But we take your suggestions into account in our revision.

4) The introduction includes a broad overview but does not emphasize the important role of the halocline, the salt-stratification that allows a vertical temperature inversion (e.g., lines 59-65). Also, the introduction does not highlight differences between coastal and open ocean polynyas.

Thanks for the comments. In our introduction, we emphasize the role of the pycnocline in relation to the thermal barrier, which is why we focus on the thermocline rather than the halocline. We will rewrite this part to mention the important role of the halocline, but will give more emphasis to the thermohaline.

**Minor issues:**

Lines 16, 160, 163, 187, etc. You switch between present and past tense, but maybe you should stick to using present tense consistently throughout the text.

OK, we will check all tenses throughout the manuscript to check for consistency and will stick to the present tense.

Line 95 and others. You vaguely talk about "physical properties" when you can be specific that you mean density.

OK, we will check over the full manuscript and we will be more specific when we talk about physical properties.

Line 64, Line 368 and other instances – you talk about "production of sensible heat" when you mean "transport" and "release"

OK, we will change it.

Line 100 "providing" –> "facilitating"

OK, we will change it.

Line 114 "within" –> "during"

OK, we will change it.

Line 119 You do not have to keep the reader waiting. Briefly state what we should expect.

OK, we will add a sentence to conclude that thermobaric convection in an important driver of stability and exchange of fluxes in the thermal barrier.

Line 55 "by associated Ekman transport" – awkward phrasing

OK, we will change it.

176 "near to the surface" –> "near the surface"

OK, we will change it.

Line 200. Diapycnal "diffusivity" is not "a process." Diffusion is a process, while diffusivity is an inherent characteristic of the system.

OK, we will change it.

Section 3 title. Why do you refer to the following as "methodology?" It seems that you are doing an overview of theory.

This is the theory and steps we follow to quantify the variables in this work.

Lines 113, 137-140, 193-194, 202-206, 286, 289, 297, 325-326, 349-353, 394, 444, 450 – awkward or confusing phrasing

OK, we will rewrite this part.

Line 226 "quanitified as" –> "defined as"

OK, we will change it.

Line 271 expand the abbreviation ASW to explain what it stands for

OK, we will expand it.

Line 301 drop "a"

OK, we will change it.

Line 335 – if the isopycnals are steep, then there is both a lateral and a vertical component to isopycnal mixing. So I would not label diapycnal and isopycnal mixing as vertical and lateral.

The isopycnals are not steep. The isopycnals squeeze because of the steep slope of the Maud Rise in an area approx. ~100 km, to keep talking about isopycnal and diapycnal mixing.

---

## Editor Comment (EC1) · Ilker Fer (Editor) · 29 Jul 2019

Dear Dr. Mojica,

Thank you for your manuscript on the mixing during a Maud Ride polynya event.

One reviewer has been (very) delayed but ensures me that the report is on its way very soon. Therefore I leave the discussion open for a few more days. I apologize for the delay.

Unfortunately, I find major shortcomings in the methods and the approach. It is therefore crucial that you return a convincing response (to my and the reviewers' comments) that demonstrates how you will satisfactorily address the issues raised. I regret to say that I would not recommend you put too much effort into preparing a revised manuscript

before I make a decision based on your final response in open discussion.

1. The polynya event reported has been presented and discussed in several recent papers in high profile journals, none of which were cited or discussed: Cheon and Gordon Scientific Reports (2019); Jena et al. GRL 2019 paper, https://doi.org/10.1029/2018GL081482; Campbell et al Nature 2019 paper (https://doi.org/10.1038/s41586-019-1294-0). I understand perhaps these papers might not have been available when you were preparing your manuscript; however, now that they are, we cannot be ignorant of the new science.

These are key works on the same event you are analyzing. In the light of these recent papers that explain the polynya formation and maintenance processes, your claims in the abstract (li 13-15), introduction (li 108-109) seem too strong ("...lack of information to a complete description...."). Furthermore you state (li 113): "...for the first time, in situ data, ...". This is not correct, see e.g. Cheon and Gordon 2019; Campbell et al 2019, who also used in situ data.

In summing up my main point 1, given the weakness in the methods (results and conclusions remain unconvincing, see below), I cannot find a new contribution in your paper on the description of the polynya event.

2. There're 3 approaches in the paper: 1) vertical mixing from Thorpe scale analyses of in situ data, 2) lateral mixing inferred from (u,v) fields of a 1/12deg resolution model, 3) heat fluxes from a double-diffusion parameterization. First of all, (1) is highly uncertain with the data at hand. Without a clear presentation of some individual profiles and Thorpe scale analysis, and a discussion of uncertainty, these Krho estimates are not convincing or acceptable. (2) is worked out from model fields in which eddy fluxes are parameterized. Given the parameterizations employed in a model, I am not convinced that the complex physics you are describing can be supported with this approach. At least a thorough discussion of caveats is needed. (3) is not meaningful at all in this system. It results in (double diffusive) heat fluxes close to nil, in a system where you

claim vertical mixing and convection is important. Most contribution to vertical heat fluxes would be turbulent and vertical entrainment during convection is likely dominant.

Overall there is also a serious disconnect between the approaches 1 to 3 above. And the results are uncertain and inconclusive.

Minor comments / clarifications:

title: "Ocean Mixing"- mixing in only indirectly inferred from coarse resolution data, and I am sorry to say, unconvincingly.

abstract, li 16-18: the study did not convincingly present processes of exchange of energy. The three relevant factors, are these shown to contribute to the energy flux, as claimed?

Sect 2.1, in situ data: please tell use how often the floats profile. And what is the sampling rate of C/T/P, the vertical profiling speed and the effective vertical resolution of the data? Is it coarser than the accuracy of 2.5 dbar? How many profiles are analyzed in total, in each region? What is a noise estimate of eddy diffusivity from the Thorpe scale analysis for a typical stratification profile?

Sect. 2.2: HYCOM: You're using (u,v) fields from 1/12deg resolution HYCOM to infer lateral fluxes. Eddy fluxes are parameterized in such models (I think). This is not described. I am not convinced these lateral fluxes from the model field will provide a description on the physics you're after. Did you consider using the float data to infer lateral diffusivity?

Sect 3. Ro number does not fit to ocean mixing section. Perhaps move/integrate to li 306 where it is used. After introducing Ro, you proceed to Krho which is very confusing and not well motivated.

li 202: Please reconsider revising "ultimately determines the variability in energy between isopycnals". Perhaps "ultimately determines the vertical stratification in the water column"?

[Figure]

please delete the equation for \detal_T in Eq(3), and introduce it in text simply as (a version of) "...is the Thorpe displacement, the vertical distance needed to move the water parcel from the observed profile to the gravitationally stable, sorted profile".

Li 214-217: This description and comparison of RMS values are very unclear. Please clarify. The maximum of RMS of 0.5 – what is it referring to?

Li 221: Start a new parag for isopycnal diffusion. In the Cole reference, is the name of Eric Kunze misspelled? You are using 1-year time averaging. The motivation for this choice or sensitivity thereof is not discussed. Seasonal variability will be interpreted as eddies.

Li 253: please insert "diffusive" before heat flux

Li 262: replace "diffusion convection" with "double-diffusion" processes

Li 273: "We identify a remarkable change of conditions between adjacent profiles confirming diffusive processes" How is this statement supported by observations? How can you rule out advection? Also the temporal sampling (e.g. number of profiles per months) is coarse (not stated) and there is a lot of interpolation (krigging?) in the figure presented.

Li 298: "below the thermohaline", you mean below the thermocline?

Li 316-330: Here I note several speculations (e.g., salinity increment from brine rejection, occurrence of diapycnal and isopycnal mixing, change in thermal barrier and energy reservoir, trigger vertical and lateral mixing etc.). Most statements remain descriptive or speculative with no attempt of quantification.

Li 417: kh does not represent lateral energy

---

## Referee Comment (RC2) · Anonymous Referee #2 · 30 Jul 2019

This is an interesting paper, one that I enjoyed reading. In general, I find that the authors have explored some new ground with the recent Weddell Sea polynya, and I believe that this paper could eventually be publishable. On the other hand, I do have some specific comments, enumerated below (some more serious than others), that I hope can be used to improve this paper.

Line 33: Should be 'Turner' instead of 'Tuner'.

Line 37: The cyclonic circulation, generated mainly by the wind stress curl, does not produce the upwelling alluded to here that is said to be due to the large-scale overturning. The overturning is presumably due to convective processes caused by vertical instabilities generated by a dense surface layer. There are cyclonic circulations driven by the wind in many places in the world ocean, but deep convective overturning doesn't

occur in most of them.

Line 113 and elsewhere: The authors state that this is 'the first time' that polynya dynamics have been characterized using *in situ* data. This is clearly untrue, as a paper published in *Nature* (June 10, 2019; volume 570, pp. 219-225) dealt with many of the same issues raised in the paper under review here. It is possible that the authors do not like the *Nature* paper or disagree with its conclusions. But it is highly misleading and not even intellectually honest not to even mention the paper in the references. Like it or not, that paper went through a rigorous review process and was published in a major journal, suggesting that the paper likely has some meritorious elements. The authors should at least acknowledge the paper and take issue with whatever parts of it they don't agree with. It is worth noting that the Nature paper used much of the same data (the SOCCOM floats) that are used in this paper.

Line 185: I doubt that HYCOM does much data assimilation in the winter, since there are no real-time data to assimilate. Thus, while the correlation of model and data might be reasonable in the summer, it is unknown how well the model does in the winter, since there is no baseline for comparison. Since the polynya occurred in late winter, it is hard to trust the model results too much.

Line 256 (equation 9) and line 301: This formulation of $F_H$ is reasonable if there is no shear to the velocity field. However, this idea is based on homogeneous turbulence, and if there is shear this formulation it won't work unless the shear is very weak. How weak? It is unknown, but the authors should attempt to estimate how weak it can be for this to be a useful parameterization.

Line 309: I believe that the authors mean $\sigma$ (with subscripts 2015 and 2017) instead of $\rho$ here.

Lines 318 and 328: The use of the conditional 'could' here sounds like pure speculation. Can this be quantified a bit more?

Line 366: The spread in the estimated values of $k_\rho$ is so large that the values hardly constrain anything; most of the global ocean above the thermocline would fall somewhere in this range.

Lines 428-434: This seems like speculation, little else.

---

## Author Comment (AC2) · 13 Sep 2019

Author's response to the editor's comments on Manuscript OS-2019-41 'Characterization of Ocean Mixing and Dynamics during the 2017 Maud Rise Polynya Event' by Mojica et al.

The authors would like to thank the editor for evaluating our manuscript and for the suggestions, which have helped to improve its clarity and quality. Our point-wise response is detailed below in blue. Iker Fer (Editor) Dear Dr. Mojica, Thank you for your manuscript on the mixing during a Maud Ride polynya event. One reviewer has been (very) delayed but ensures me that the report is on its way very soon. Therefore I leave the discussion open for a few more

days. I apologize for the delay. Unfortunately, I find major shortcomings in the methods and the approach. It is therefore crucial that you return a convincing response (to my and the reviewers' comments) that demonstrates how you will satisfactorily address the issues raised. I regret to say that I would not recommend you put too much effort into preparing a revised manuscript before I make a decision based on your final response in open discussion.

Thank you for giving us the chance to address the lack of clarity in the method and the approach. Kindly find below our point-by-point answers to the comments that were raised. In the updated version of the manuscript currently under preparation, we rigorously address all the questions regarding the methods and approaches that you and the reviewer have pointed out.

1. The polynya event reported has been presented and discussed in several recent papers in high profile journals, none of which were cited or discussed: Cheon and Gordon Scientific Reports (2019); Jena et al. GRL 2019 paper, https://doi.org/10.1029/2018GL081482; Campbell et al Nature 2019 paper (https://doi.org/10.1038/s41586-019-1294-0). I understand perhaps these papers might not have been available when you were preparing your manuscript; however, now that they are, we cannot be ignorant of the new science.

These are key works on the same event you are analyzing. In the light of these recent papers that explain the polynya formation and maintenance processes, your claims in the abstract (li 13-15), introduction (li 108-109) seem too strong ("...lack of information to a complete description...."). Furthermore, you state (li 113): "...for the first time, in situ data, ...". This is not correct, see e.g. Cheon and Gordon 2019; Campbell et al 2019, who also used in situ data.

In summing up my main point 1, given the weakness in the methods (results and conclusions remain unconvincing, see below), I cannot find a new contribution in your paper on the description of the polynya event.

As you already mentioned, during the preparation and initial submission of our manuscript none of those papers was published. Now that they are, we will certainly include their findings when addressing the state-of-the-science in the introduction, and we will state clearly the additional contribution our study brings in relation to this new work. For instance, we have found similar results to Cheon and Gordon 2019 and Campbell et al., 2019 with regards to convective mixing. While these two studies describe the ocean and atmospheric interaction over Maud Rise during the 2017 Polynya event, we focus on the ocean preconditioning during the years leading up to the occurrence of the Polynya. To do this, we quantified the mixing rates to create a mixing map over Maud Rise. We then compared mixing rates over Maud Rise with those elsewhere to highlight and describe the role of the bathymetry.

We have also deleted the statements in lines 13-15, 108-109, 113, to acknowledge the manuscripts published recently.

2. There're 3 approaches in the paper: 1) vertical mixing from Thorpe scale analyses of in situ data, 2) lateral mixing inferred from (u,v) fields of a 1/12deg resolution model, 3) heat fluxes from a double-diffusion parameterization. First of all, (1) is highly uncertain with the data at hand. Without a clear presentation of some individual profiles and Thorpe scale analysis, and a discussion of uncertainty, these Krho estimates are not convincing or acceptable. (2) is worked out from model fields in which eddy fluxes are parameterized. Given the parameterizations employed in a model, I am not convinced that the complex physics you are describing can be supported with this approach. At least a thorough discussion of caveats is needed. (3) is not meaningful at all in this system. It results in (double diffusive) heat fluxes close to nil, in a system where you claim vertical mixing and convection is important. Most contribution to vertical heat fluxes would be turbulent and vertical entrainment during convection is likely dominant.

Overall there is also a serious disconnect between the approaches 1 to 3 above. And the results are uncertain and inconclusive.

Regarding the three approaches:

The Thorpe scale is used to identify the vertical overturning scale from fine-scale density profiles (Garget and Garner, 2008). When applying careful data analysis and processing such as we describe in another comment below (see the comment for Lines 214-217 - RMS), the Thorpe scale is a solid and widely used method to quantify diffusivity (e.g. Park et al., 2014). The Thorpe scale, calculated from the SOCCOM profiles, provides a robust approach to identify convection-driven changes in weakly stratified regions. We analyze these variations to the North of, and over, Maud Rise. The convection-led mixing processes can be seen clearly in the Krho estimations, which identify when thermal barrier reach shallow waters increasing the ventilation at the surface (Figure 3). We also include Thorpe scale profiles as part of Figure 3 (see below), and analyze these in section 4.2.1 to support our diapycnal diffusivity calculation.

Figure 1. Thorpe scale, float *68 (over Maud Rise) and *97 (North of Maud Rise). (Blue rectangle) Note the remarkable decrease at the thermal barrier depth for float *97 compared to the overturning at Maud Rise. Figure to be included in the manuscript as part of Figure 3.

We agree with you regarding the current inability of the model to resolve such high-resolution processes, and we have now included a description to highlight the limitations of the model. As you mentioned, we used velocity fields from HYCOM to infer lateral fluxes. However, in light of the recent literature and given the fact that lateral fluxes play a very small role in polynya preconditioning, we will shorten the section on this topic and reorganize subsections 2.2 and 4.2.2 to emphasize our analysis of the vertical mixing processes.

The values of heat fluxes are small because we only quantified the range across the interface of the thermal barrier. We now include the values across the water column $(0 - 1000$ m, $\sim 200$ W m-2). In this way, we can relate the flux variability to the convective mixing processes we quantified previously, thus connecting approaches 1

and 3 together. We include more information on this in table 1 and describe the new heat fluxes in section 4.2.3.

Following the previous statements, we will connect the 3 approaches through figures 2 and 3 where we see the effects of the convective mixing in shallow water (low stratification, small variability in salinity), and thermobaric convection below the thermal barrier (plumes on the thermobaric coefficients and variability in the diapycnal diffusivity). We will make a short statement on the low contribution of the lateral fluxes in order to give a full description of the oceanic processes that provide preconditioning for the Maud Rise Polynya.

Minor comments / clarifications:

title: "Ocean Mixing"- mixing in only indirectly inferred from coarse resolution data, and I am sorry to say, unconvincingly. We will change the manuscript title to "Characterization of Ocean Convective mixing during the 2017 Maud Rise Polynya Event". In this way, we clarify our focus subject as the assessment of the ocean conditions that make Maud Rise susceptible to a polynya opening.

abstract, line 16-18: the study did not convincingly present processes of exchange of energy. The three relevant factors, are these shown to contribute to the energy flux, as claimed?

In our original figures 3 and 6, we presented a temporal evolution of the thermal barrier during the polynya event. The three factors you mentioned contribute in part to the energy flux. We will indicate in the temporal series the decreasing surface buoyancy that produces convection. This generates instabilities at the thermal barrier which, to maintain a balance, supplies energy via thermobaric processes throughout the water column. In the updated version, we will include a statistical analysis in section 4, to quantify how these variables changed in 2017 compared to previous years, and describe how the decreasing buoyancy broke the thermal barrier.

[Figure]

Line 16-18 rewritten 'The results reveal that the incidence of convective mixing and thermobaric convection over the Maud Rise drives the exchange of energy in the water column'.

Sect 2.1, in situ data: please tell us how often the floats profile. And what is the sampling rate of C/T/P, the vertical profiling speed and the effective vertical resolution of the data? Is it coarser than the accuracy of 2.5 dbar? How many profiles are analyzed in total, in each region? What is a noise estimate of eddy diffusivity from the Thorpe scale analysis for a typical stratification profile?

Float profiles: ~10 days. Sampling data binned into 2dbar intervals. Measurement accuracy for temperature: 0.005°C, salinity 0.01. Speed of ascent: 8-10 cm/sec. On average, we consider 140 profiles for each float. We have included this information in Section 2, line 144-147.

Sect. 2.2: HYCOM: You're using (u,v) fields from 1/12deg resolution HYCOM to infer lateral fluxes. Eddy fluxes are parameterized in such models (I think). This is not described. I am not convinced these lateral fluxes from the model field will provide a description on the physics you're after. Did you consider using the float data to infer lateral diffusivity?

Please see previous reply about the use of HYCOM.

Sect 3. Ro number does not fit to ocean mixing section. Perhaps move/integrate to li 306 where it is used. After introducing Ro, you proceed to Krho which is very confusing and not well motivated.

Thanks for the suggestion. We moved and clarified the inclusion of the Ro number in section 4.1. Line 299-302. "To assess the regime of the system, we estimate the Rossby number ($R_O=U/fL$, where U is the velocity scale, f is the Coriolis parameter, and L is the horizontal length scale). Considering the MR length scale, Ro = 0.02 relates a weak inflow, and the conditions of weak stratification, and upwelling, conducive

to the formation of a Taylor cap (Ou, 1991; Alverson and Owens, 1996)."

li 202: Please reconsider revising "ultimately determines the variability in energy between isopycnals". Perhaps "ultimately determines the vertical stratification in the water column"? please delete the equation for ndetal_T in Eq (3), and introduce it in text simply as (a version of) "...is the Thorpe displacement, the vertical distance needed to move the water parcel from the observed profile to the gravitationally stable, sorted profile".

Thanks for the suggestion. We rewrote line 202 following your suggestion, and rewrote the equations; including our original equation 3, as a simple text version (line 204-205).

Li 214-217: This description and comparison of RMS values are very unclear. Please clarify. The maximum of RMS of 0.5 – what is it referring to?

We calculated the RMS values of the sorting potential density. Dividing by RMS Thorpe fluctuation scales the T and S deviation to the density amplitude of the suspected overturn. The resultant ratios were scored on a scale from 0 - 1, according to the strength of the T-S relationship. Scores below 0.5 were assigned to regions that would be discarded (e.g. regions with loops), in this way, we have T-S relationships that are robust enough to be regarded as a signature of overturning motion. We have included this information in new lines 214-217.

Li 221: Start a new parag for isopycnal diffusion. In the Cole reference, is the name of Eric Kunze misspelled? You are using 1-year time averaging. The motivation for this choice or sensitivity thereof is not discussed. Seasonal variability will be interpreted as eddies.

We started a new paragraph for isopycnal diffusion from line 221. We reviewed and corrected the reference. We initially chose 1-year time (Line 232) averaging to get the overall picture of annual salinity evolution and to understand why this process did not fully develop in previous years. We will include a seasonal analysis to identify the

fluxes and avoid any misunderstanding with possible eddies. This will form part of section 4.2.2 lines 386-399 in the updated manuscript.

Li 253: please insert "diffusive" before heat flux

Done, line rewritten.

Li 262: replace "diffusion convection" with "double-diffusion" processes

Done, line rewritten.

Li 273: "We identify a remarkable change of conditions between adjacent profiles confirming diffusive processes" How is this statement supported by observations? How can you rule out advection? Also the temporal sampling (e.g. number of profiles per months) is coarse (not stated) and there is a lot of interpolation (krigging?) in the figure presented.

We don't rule out advection. The high salinity and small variability between consecutive profiles (Fig. 2) suggest that advection makes a small contribution. A relevant process for polynya precondition is brine rejection during sea-ice formation, which creates the conditions for convective mixing. Buoyancy profiles (Figure 2), show a destabilized upper ocean due to the increased salinity, as we presented in line 318-320. During the sea-ice melting period, the system tries to balance the subsequent freshening by creating a weak stratification that allows convective mixing in the surface water layer. We include this information on advection in lines 322-325. The observations (temporal sampling) allow us to identify the small salinity variation during the polynya event. In figure 2, there is no interpolation, which is why there is missing data in the regions where the SOCCOM buoys cannot reach the surface because of the presence of ice. We include a mark to identify each of the individual profiles in figures 2, 3, 6, and S1.

Li 298: "below the thermohaline", you mean below the thermocline?

Yes, changed.

Li 316-330: Here I note several speculations (e.g., salinity increment from brine rejection, occurrence of diapycnal and isopycnal mixing, change in thermal barrier and energy reservoir, trigger vertical and lateral mixing etc.). Most statements remain descriptive or speculative with no attempt of quantification.

We included a description and quantification of the variability of the surface salinity in lines 316-330 that will help to clarify why we ruled out advection, as described in the previous comment. In addition, we included analysis and statements that support our approach for diapycnal and isopycnal mixing.

Li 417: kh does not represent lateral energy Yes, kh represents the horizontal diffusivity; we changed this in line 417.

Please also note the supplement to this comment:
https://www.ocean-sci-discuss.net/os-2019-41/os-2019-41-AC2-supplement.pdf

[Figure]

**Fig. 1.**

---

## Author Comment (AC3) · 13 Sep 2019

Authors response to the reviewer's comments on Manuscript OS-2019-41 'Characterization of Ocean Mixing and Dynamics during the 2017 Maud Rise Polynya Event' by Mojica et al.

The authors would like to thank the reviewer for evaluating our manuscript and for the suggestions which helped to improve the clarity and the quality of the paper. Our point-wise response is detailed below in blue. Anonymous Referee # 2 This is an interesting paper, one that I enjoyed reading. In general, I find that the authors have explored some new ground with the recent Weddell Sea polynya, and I believe that this paper could eventually be publishable. On the other

hand, I do have some specific comments, enumerated below (some more serious than others), that I hope can be used to improve this paper.

Thank you for all the comments and suggestions. We respond to them below point-by-point.

Line 33: Should be 'Turner' instead of 'Tuner'.

Yes, changed

Line 37: The cyclonic circulation, generated mainly by the wind stress curl, does not produce the upwelling alluded to here that is said to be due to the large-scale over-turning. The overturning is presumably due to convective processes caused by vertical instabilities generated by a dense surface layer. There are cyclonic circulations driven by the wind in many places in the world ocean, but deep convective overturning doesn't occur in most of them.

We agree. The upwelling concentrates warm and salty Weddell Deep Water close to the surface, thereby affecting the stratification. We clarify this idea about upwelling and include your statement about the overturning due to convective processes in line 37, following the literature (Campbell et al., 2019).

Line 113 and elsewhere: The authors state that this is 'the first time' that polynya dynamics have been characterized using in situ data. This is clearly untrue, as a paper published in Nature (June 10, 2019; volume 570, pp. 219-225) dealt with many of the same issues raised in the paper under review here. It is possible that the authors do not like the Nature paper or disagree with its conclusions. But it is highly misleading and not even intellectually honest not to even mention the paper in the references. Like it or not, that paper went through a rigorous review process and was published in a major journal, suggesting that the paper likely has some meritorious elements. The authors should at least acknowledge the paper and take issue with whatever parts of it they don't agree with. It is worth noting that the Nature paper used much of the same

data (the SOCCOM floats) that are used in this paper.

During the preparation and initial submission of our manuscript, none of those papers was published. Now that they are, we will certainly include their findings when addressing the state-of-the-science in the introduction and we will state clearly the additional contribution our study brings in relation to this new work. For instance, we have found similar results to Cheon and Gordon 2019 and Campbell et al., 2019 with regards to convective mixing. While these two studies describe the ocean and atmospheric interaction over Maud Rise during the 2017 polynya event, we focus on the ocean preconditioning during the years leading up to the occurrence of the Polynya. To do this, we quantified the mixing rates to create a mixing map over Maud Rise. We then compared mixing rates over Maud Rise with those elsewhere to highlight and describe the role of the bathymetry.

We have also deleted the statements on lines 13-15, 108-109, 113, to acknowledge the manuscripts published recently.

Line 185: I doubt that HYCOM does much data assimilation in the winter, since there are no real-time data to assimilate. Thus, while the correlation of model and data might be reasonable in the summer, it is unknown how well the model does in the winter, since there is no baseline for comparison. Since the polynya occurred in late winter, it is hard to trust the model results too much.

We agree that there is a lack of in situ vertical profiles during winter in this region - however, the model still assimilates surface altimeter data. In the absence of vertical profiles, HYCOM uses MODAS (Modular Ocean Data Assimilation System) to generate synthetic profiles (T and S) that are consistent with the along-track altimeter SSHA. Where there is no data to assimilate, the model uses conservative advection routines and conserves salt.

However, in light of the recently published papers and given the fact that the values of lateral fluxes are negligible, we decided to shorten the section using HYCOM outputs.

We reorganized subsection 2.2 and 4.2.2 to reflect this, and we emphasized instead the vertical processes, which appear to be the most relevant.

Line 256 (equation 9) and line 301: This formulation of FH is reasonable if there is no shear to the velocity field. However, this idea is based on homogeneous turbulence, and if there is shear this formulation won't work unless the shear is very weak. How weak? It is unknown, but the authors should attempt to estimate how weak it can be for this to be a useful parameterization.

As seen in the vertical distribution of current velocities (figure 4), shear in this area is weak, because the velocity field has a small velocity gradient over the water column. Velocity values decrease with depth by 0.04 cms-1. We discussed these values in section 4.2.2 lines 373-381. Therefore, our representation of FH is appropriate. We will note on line 256 that there is negligible shear to the velocity field.

Line 309: I believe that the authors mean _ (with subscripts 2015 and 2017) instead of _ here.

Yes, we changed the writing format to emphasize the subscripts.

Lines 318 and 328: The use of the conditional 'could' here sounds like pure speculation. Can this be quantified a bit more?

Absolutely, we included values to illustrate the change in salinity and temperature in the surface waters between consecutive profiles previous to and during the Polynya event (average salinity variation $\sim$ 0.01, temperature $\sim$0.02°C, and buoyancy variability of $\sim$ 0.1 cph; and salinity values of $\sim$ 0.03, temperature $\sim$ 0.1 °C and buoyancy $\sim$ 0.3 cph, respectively). This information was included in section 4.1, lines 318-320.

Line 366: The spread in the estimated values of k_ is so large that the values hardly constrain anything; most of the global ocean above the thermocline would fall somewhere in this range.

There are no previous measurements of diapycnal diffusivity at the same location,

which is why we refer to the closest records from two previously published works. We constrain the values measured by Naveria Garabato et al., 2004a to shallow waters, in this case the range changes from $\sim$3x10-4 – 1 x 10-4 m2s-1. We included this information in line 366.

Lines 428-434: This seems like speculation, little else.

We rewrote this paragraph, comparing the heat transfer values (0.1 Wm-2 during the Polynya event to 0.04 Wm-2 in summer at the same depth) from figure 6 with the thermal and salinity composition during the same period (thermobaric coefficient 3.1 during the Polynya event compared to 2.9 in summer at the same depth). In this way, we demonstrate that these convection processes perturbed the thermal barrier and increased the ventilation, thereby helping to produce the Polynya event in 2017. We added this information in lines 428 - 434.

Please also note the supplement to this comment:
https://www.ocean-sci-discuss.net/os-2019-41/os-2019-41-AC3-supplement.pdf